

# Fluvio-deltaic record of increased sediment transport during the Middle Eocene Climatic Optimum (MECO), Southern Pyrenees, Spain

Sabí Peris Cabré[1, 2], Luis Valero[1], Jorge E. Spangenberg[3], Andreu Vinyoles[4], Jean Verité[1, 5], Thierry Adatte[6], Maxime Tremblin[1], Stephen Watkins[1], Nikhil Sharma[1], Miguel Garcés[4], Cai Puigdefàbregas[1], Sébastien Castelltort[2]

[1]Département des Sciences de la Terre, Université de Genève, Genève, 1205, Switzerland
[2]Departament de Geologia, Universitat Autònoma de Barcelona, Cerdanyola del Vallès, 08193, Spain
[3]Institute of Earth Surface Dynamics (IDYST), University of Lausanne, Géopolis, Lausanne, 1015, Switzerland
[4]Departament de Dinàmica de la Terra i l'Oceà, Facultat de Ciències de la Terra, Barcelona, 08028, Spain
[5]Départment des Geosciences, Université de Rennes, Rennes, UMR 6118, France
[6]Institute of Earth Sciences (ISTE), University of Lausanne, Géopolis, Lausanne, 1015, Switzerland

*Correspondence to*: Sabí Peris Cabré (sabi.peris@uab.cat)

**Abstract.** The early Cenozoic marine sedimentary record is punctuated by several brief episodes (< 200 kyr) of abrupt global warming, called hyperthermals, that have disturbed ocean life and water physicochemistry. Moreover, recent studies of fluvial-deltaic systems, for instance at the Palaeocene-Eocene Thermal Maximum, revealed that these hyperthermals also impacted the hydrologic cycle, triggering an increase of erosion and sediment transport at the Earth's surface. Contrary to the early Cenozoic hyperthermals, the Middle Eocene Climatic Optimum (MECO), lasting from 40.5 to 40.0 Ma, constitutes an event of gradual warming that left a highly variable carbon isotopic signature and for which little data exist about its impact on Earth surface systems. In the South-Pyrenean Foreland Basin (SPFB), an episode of prominent deltaic progradation (Belsué-Atarés and Escanilla formations) in the middle Bartonian has been usually associated to increased Pyrenean tectonic activity, but recent magnetostratigraphic data suggest a possible coincidence between the progradation and the MECO warming period. To test this hypothesis, we measured the stable isotope composition of carbonates and organic matter of 257 samples in two sections of SPFB fluvial-deltaic successions covering the different phases of the MECO and already dated with magnetostratigraphy. We find a negative shift in $\delta^{18}O_{carb}$ and an unclear signal in $\delta^{13}C_{carb}$ around the transition from magnetic Chron C18r to Chron C17r (middle Bartonian). These results allow, by correlation with reference sections in the Atlantic and Tethys, to identify the MECO and document its coincident relationship with the Belsué-Atarès fluvial-deltaic progradation. Despite its long duration and a more gradual temperature rise, the MECO in the South Pyrenean Foreland Basin may have led, like lower Cenozoic hyperthermals, to an increase of erosion and sediment transport that is manifested in the sedimentary record. The new data support the hypothesis of a more important hydrological response to the MECO than previously thought in mid latitude environments, including those around the Tethys.



## 1 Introduction

The Middle Eocene Climatic Optimum (MECO) is a global warming event that occurred during the Bartonian (ca. 40.5 – 40.0
Ma), and which briefly reversed the longer-term cooling trend of the middle to upper Eocene (Fig. 1; Arimoto et al., 2020; Bijl
et al., 2010; Bohaty and Zachos, 2003; Bohaty et al., 2009; Bosboom et al., 2014; Galazzo et al., 2014; Henehan et al., 2020;
Edgar et al., 2010, 2020; Giorgini et al., 2019; Jovane et al., 2007; Mulch et al., 2015; Sluijs et al., 2013; Sppoforth et al., 2010;
Pälike et al., 2012; van der Boon et al., 2020). Marine bulk and benthic oxygen isotope composition ($\delta\,^{18}$O) both show a
negative excursion of -1.5 ‰ over the event, interpreted as a gradual global warming of 3 to 6°C (Bohaty et al., 2009). The
evolution of carbon isotope composition $\delta^{13}$C, in contrast, and unlike earlier hyperthermals of the Cenozoic (e.g., Palaeocene-
Eocene Thermal Maximum PETM, Eocene Thermal Maximum ETM 2 among others), differs from site to site, showing
opposite patterns between hemispheres and displaying a carbon isotope excursion (CIE) in some but not all marine records
(Bohaty et al., 2009; Henehan et al., 2020; Westerhold and Röhl, 2013). This CIE suggests a raise in atmospheric partial
pressure of carbon dioxide (pCO2) during the warming peak (Henehan et al., 2020; Bijl et al., 2010), and numerous potential
CO2 sources have been proposed. Among them, a prolonged pulse of metamorphic decarbonization possibly linked with
Himalayan collision at that time (Bijl et al., 2010; Bohaty et al., 2009, Bouilhol et al., 2013, Sternai et al., 2020), an increase
of volcanism (van der Boon et al. 2020), or lower continental weatherability (van der Ploeg et al. 2018). However, the pCO2
record remains ambiguous and difficult to link in a straightforward way to a rapid injection of exogenous carbon during the
MECO (e.g., Henehan et al., 2020). In addition, regardless of the CO2 sources involved, the MECO coincides with a 2.4-Myr
very long-term eccentricity cycle (2.4 My) minima (Westerhold & Röhl, 2013), which suggests a possible orbital trigger
(Westerhold & Röhl, 2013, Henehan et al. 2020). Therefore, considering the unresolved MECO driving mechanism(s), and
how the Earth system responded to this carbon cycle perturbation, the MECO is currently considered as a key problem in
paleoclimate research, holding keys about our understanding of the global carbon cycle in the larger context of the solid and
fluid Earth interactions (Sluijs et al., 2013; Henehan et al., 2020; Sternai et al., 2020).

Current data converge towards the view that, during the MECO, surface and deep oceanic waters suffered a gradual and
uniform warming between 3 to 6°C for all latitudes (Arimoto et al., 2020; Bijl et al., 2010; Bohaty et al., 2009; Rivero-Cuesta
et al., 2019). Moreover, deep-sea carbonates were affected either by very reduced carbonate accumulation rates or even by
dissolution, suggesting broad acidification of sea-bottom waters, involving an estimated *ca* 1-km shoaling of the carbonate
compensation depth (CCD; Henehan et al., 2020; Cornaggia et al. 2020; Pälike et al. 2012; Arimoto et al., 2020). Finally, this
significant environmental perturbation seems to have caused widespread ocean stratification and eutrophic conditions, starving
the benthic foraminiferal communities during the climax of MECO warmth (Galazzo et al., 2014; Arimoto et al. 2020;
Cramwinckel et al. 2019).





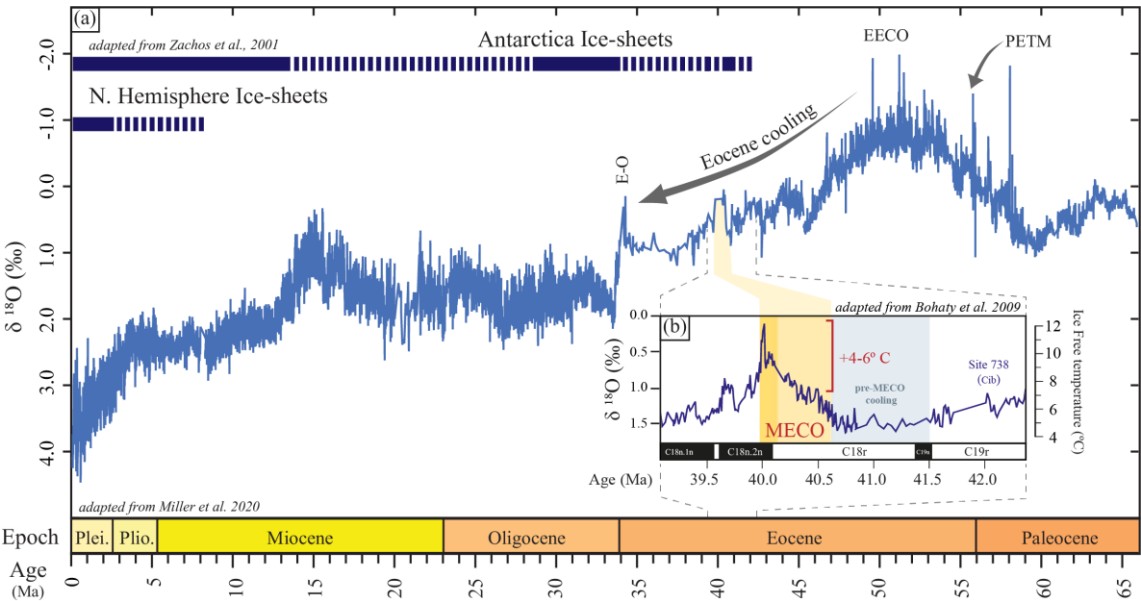

**Figure 1: (a) Cenozoic δ¹⁸O values compilation from the Pacific Ocean, compiled in Miller et al. (2020). The continuous blue bar represents permanent ice sheet presence, and the discontinuous blue bar represents ephemeral ice sheet, modified from Zachos et al. 2001. Main climate events, EECO: Early Eocene Climatic Optimum; PETM: Palaeocene Eocene Thermal Maximum, MECO: Middle Eocene Climatic Optimum, E-O: Eocene Oligocene transition. (b) Carbonate δ¹⁸O values from site 702, by Bohaty et al., 2009. General climatic context of the Middle Eocene Climatic Optimum. The MECO "event", in yellow from ca 40.5 Ma to 40 Ma in the inset, is considered the last "hyperthermal" of the Eocene, immediately preceding the shift to genuine Antarctic glaciation and the ice-house world of the Oligocene.**

In contrast to the oceanic realm, the expression of the MECO in non-marine records remains scarce and variable. Mulch *et al.* (2015) suggested a boost of precipitation in the North American plateau derived from low δ¹⁸O values, while Bosboom *et al.* (2014) documented a shift towards arid conditions in the Tarim Basin with a reduction in fern palynomorphs. Such drying trend in central Asia is opposite to the Neo-Tethys ocean dynamic, where a greater burial of organic matter (OM) immediately following the MECO may have been caused by increased nutrients runoff due to an enhanced hydrological cycle during the warm period (Galazzo et al., 2014; Giorgioni et al., 2019; Spofforth et al., 2010). These studies raise the question of the response of weathering, erosion, and sediment transport in terrestrial systems to global warming as it has been posed also for other hyperthermals recently (e.g., Chen et al., 2018; Foreman et al., 2012, 2017; Honegger et al., 2020). This highlights the need for further documentation of the clastic sedimentary successions that temporally cover single and long-term climate crises (i.e. Early Eocene Climatic Optimum, Palaeocene Eocene Thermal Maximum, etc Fig. 1).

In this work, we aim to understand the effects of the MECO on surface systems by exploring the interface between ocean and continent. The shallow marine settings, very sensitive to sea level changes and sediment supply, potentially provides a unique perspective of the hydrological response to climate change in the continental domain, as well as geochemical and isotopic evolution in the marine domain. We focus on two separated deltaic successions in the southern (Belsué locality, B) and northern (Yebra de Basa locality, YB) margins of the Jaca basin in the South-Pyrenean foreland basin (SPFB; Fig. 1). The successions are characterized by excellent exposure and have already been dated thanks to high-resolution magnetostratigraphy. Both





sections reveal progradation of deltaic and fluvial systems coeval with the magnetic reversal occurring at chrons C18r and

C18n.2n, near or at the zenith of MECO warmth (Edgar et al., 2010, 2020; Garcés et al., 2014; Vinyoles et al., 2021). We

generated new high-resolution profiles of $\delta^{13}C$ and $\delta^{18}O$, XRF, clays, and Rock Eval, across the Chron C18r-C18n.2n reversal,

in order to identify geochemical changes associated with the MECO onset and its recovery and test the possible causative links

between progradation and the MECO perturbation. Finally, we discuss the sedimentary evolution of both sections to understand

landscape response during the MECO, and we explore the significance of its identification in the SPFB and the impact of

climate shifts in source to sink systems as recorded at the continental to ocean interface.

## 2 Geological Setting


The Pyrenees are a nearly E-W trending mountain belt formed by the collision of the Iberian and European plates from Late

Cretaceous (Santonian) to the Early Miocene (Muñoz, 1992; Roure et al., 1989; Teixell, 1998; Vergés et al. 2002). The south

Pyrenean zone is composed of an imbricate system of synorogenic thrusted cover sheets propagating southwards, detached

above the Triassic evaporites (Labaume and Teixell, 2018; Lagabrielle et al., 2010; Mochales et al., 2012; Pueyo, et al., 2002;

Teixell et al., 2016, 2018). Among them, the emplacement of the South-Central Unit (SCU) by early Eocene resulted in the

partition of the South Pyrenean Basin (SPFB) and the development of an E-W elongated deep basin draining west towards the

Atlantic Ocean (Mochales et al., 2012; Muñoz et al., 2018; Puigdefàbregas, 1975, Puigdefàbregas and Souquet, 1986; Séguret,

1972). Due to the westward propagation of deformation during the middle Eocene and the differential velocity of the thrust

sheets, oblique thrusted anticlines developed at the south-western termination of the SCU (Muñoz et al., 2013). These thrusts

caused the fragmentation and piggy-back transport of a wider foreland region, which included from east to west: the Tremp-

Graus, Ainsa, and eventually the Jaca basins (Muñoz et al., 2018; Fig. 2).

The Tremp-Jaca basin (TJB) preserves an outstandingly exposed complete source-to-sink system during MECO times. In the

middle Eocene, the alluvial-fluvial system of Sis-Escanilla flowed down the Tremp-Graus and Ainsa basins, draining the

eroded sediments from the uplifting north-eastern Pyrenees (Beamud et al., 2003; Roigé et al., 2016; Coll et al., 2020;

Puigdefábregas, 1975, 1986). Sediments were transported westwards into the Jaca basin, forming a mixed delta-carbonate

ramp with tidal influence (Puigdefàbregas, 1975; Castelltort et al., 2003). Two main deltaic systems developed during lower

Bartonian in the southern (Belsué-Atarés) and northern (Sabiñánigo) margin of the Jaca basin, that primarily fed the distal

Hecho turbidites in the western sector (Mutti, 1977; Remacha and Fernández, 1985). Shallow marine environments are mainly

dominated by marly facies, which thanks to their high carbonate content and the relatively deep depositional environment are

suitable for geochemical purposes (Wendler, 2013). Available high-resolution magnetostratigraphy in the Pyrenean region

provides a correlation of different sections along the entire source-to-sink system (Vinyoles et al., 2021). We selected two

lower Bartonian sections, Belsué (BS) and Yebra de Basa (YB), because they present excellent exposition and are provided

with magnetostratigraphic dating, aiming to unravel the geochemical history of the two main deltaic systems coeval to the

MECO.






**Figure 2: Geological and Stratigraphic context of the study area (a) Synthetic geologic map of the Pyrenees and location of the study area in the Jaca Basin (red square). Black lines represent the main tectonic structures. Modified from Teixell (1996) and Bosch (2016). (b) Detailed geologic map of the Jaca basin and the study area (Belsué and Yebra de Basa). Modified from Puigdefàbregas (1975) and Remacha (1996). (c) Chronostratigraphy of the Ainsa and Jaca basins, showing the westwards progradation of all depositional systems. The names of the main lithostratigraphic units are represented in black. The studied sections are included in the work of Vinyoles et al., 2021 (in white) whilst the location of the geochemical analyses carried out in this paper is highlighted in red. The figure is modified from Vinyoles et al. (2021).**




The 130-m thick BS section is located within the External Sierras, east of the "Pico del Águila" anticline (Fig. 3; 42.30° N 0.37° W). This section has been extensively studied as a perfect case study for the tectonic–sedimentation relationship (Fig. 2; (Puigdefàbregas, 1975; Puigdefàbregas and Souquet, 1986; Millán et al., 1994, 2000; Castelltort et al., 2003; Huyghe et al.,

2012; Garcés et al., 2014). The lower boundary corresponds to an encrusted and ferruginous surface on top of a shallow marine bioclastic limestone (Guara Fm.), locally overlain by sandy marls rich in glauconite (Millán et al., 1994). It has been interpreted as a drowning unconformity of the Guara carbonate platform (Puigdefàbregas, 1975; Silva-Casal et al., 2019), close to the Lutetian-Bartonian boundary (Rodriguez Pintó et al., 2013). This major unconformity led to the syntectonic deposition of the Arguís marls and the Belsué sandstones, while the Gabardiella and Pico del Águila anticlines were growing. Different authors

studied the influence in the stratigraphy of local tectonic movement in Belsué-Arguís region (Lafont, 1994; Castelltort et al., 2003), concluding that local tectonics modify the stacking pattern and position of its genetic units along with different depositional environments. The entire section covers the lower Bartonian interval (Garcés et al., 2014) up to a maximum flooding surface MFS–2 by Muñoz et al. (1994; Fig. 3), which corresponds to the deepest paleobathymetry in the BS area (*ca.* 150 m, Sztràkos and Castelltort, 2001).

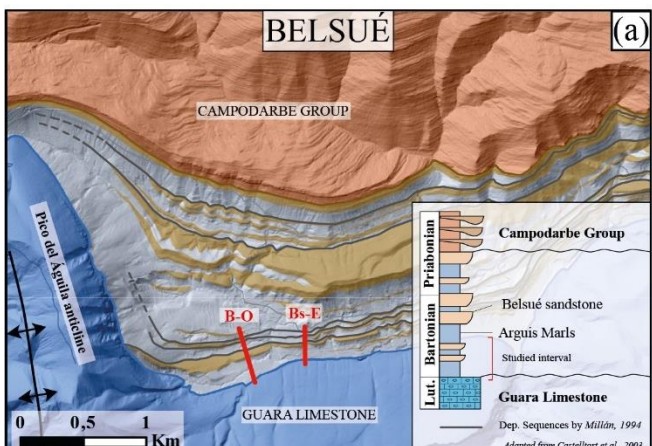
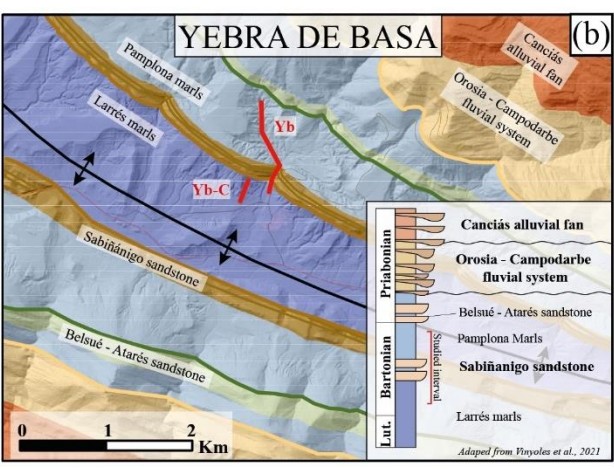


**Figure 3. Detailed geologic maps from Belsué (a) and Yebra de Basa (b), modified respectively from Puigdefàbregas (1975) and Remacha (1996). Small stratigraphic logs on the bottom right represent large-scale synthetic logs representing the different formations name, which are modified from Vinyoles et al. (2021) for Yebra de Basa and Castelltort et al. (2003) for Belsué. Red lines represent the studied sections location, and the grey lines in Belsué represent the depositional sequences defined by Millán et al. (1994).**

The 800-m YB section is located between the Basa anticline and the Santa Orosia syncline (Fig. 3; 42.49° N 0.28° W), and comprises one of the best outcropping sections for the Sabiñánigo sandstone (Puigdefàbregas, 1975; Lafont, 1994; Boya, 2018). It is composed of three subsections covering the lower Bartonian interval (Fig. 3), between the upper part of the chron C18r and the chron C18n.1r (Vinyoles et al., 2021). The first subsection is located west (Yb-C), within the Larrés marls, whose top is correlated with the base of the Vinyoles et al., (2021) magnetostratigraphic profile. From here, the next two subsections

were built following the Vinyoles et al. (2021) profile, which comprises the two deltaic levels of the Sabiñánigo sandstone and most of the overlaying Pamplona marls.





# 3 Materials and methods

## 3.1 Field and sampling

We performed a complete stratigraphic study and sampling of the lower Bartonian sections from BS and YB. The stratigraphic
thickness of these sections was measured using the Jacobs staff in the field and geometric calculations when direct
measurements were impossible. New mapping based on fieldwork and orthophotos was performed to correlate the different
subsections. The Belsué section in the southern margin is composed of two subsections (Fig. 3); the lower one (BS-E) was
sampled at a resolution of 0.5–1.0 m and the upper one (BS-O) every 3–6 m. In the northern margin, the higher sedimentation
rate in Yebra de Basa (>80 cm/kyr; Vinyoles et al., 2021) motivated a high-resolution sampling of 1–3 m at the middle part of
the section (YB-HR), and sampling each 9–15 m in the other intervals (YB-C and YB-sup).

A total of 101 samples in BS and 157 samples in YB were collected, each of them was composed by *ca.* 200 g of fine-grained
and fresh rock from below the weathering depth to avoid alteration and grain size bias (e.g. Lupker et al., 2011). These samples
were prepared for mineralogical and geochemical analyses in the laboratories of the University of Geneva and the University
of Lausanne. The sample surface was cleaned with deionized water, the weathered material was removed, and then dried at
45ºC for 2–3 days. The dried samples were crushed with a hydraulic press and powdered using an agate mill.

## 3.2 Clay Mineralogy (XRD)

The clay mineralogical assemblages of 24 representative samples per section were determined by X-ray diffractometry. The
used system was a Thermo Scientific ARL X-TRA diffractometer at the Institute of Earth Science of the University of
Lausanne (ISTE-UNIL), following the methods described by Klug and Alexander (1974), Kübler (1983, 1987) and Adatte et
al. (1996). Ground chips were mixed with deionized water (pH 7 to 8) and agitated. The carbonate fraction was removed by
treatment with 10% HCl at room temperature and then for 20 minutes or more until all carbonate was dissolved. The insoluble
residue was disaggregated (ultrasonication, 3 min), washed and centrifuged (8 times) until a neutral suspension was obtained
(pH 7–8). Different grain size fractions (<2 to 16 μm) were separated by the time settling method based on Stokes law. The
selected fraction was then pipetted onto a glass plate and air-dried at room temperature. XRD analyses of oriented clay samples
were made after air drying at room temperature at ethylene-glycol-solvated conditions. The intensities of XRD peaks (2θ,
Moore and Reynolds 1997) characteristic of each clay mineral (*e.g.,* chlorite, mica, kaolinite) were used for a semi-quantitative
estimation of the relative percent of clay minerals present in two size fractions (<2 μm and 2–16 μm).

## 3.3 Major and trace element composition (XRF)

Major and trace element concentrations from 24 representative samples per section were determined by X-ray fluorescence
(XRF) spectrometry, using a PANalytical PW2400 spectrometer from ISTE-UNIL. The major elements were analysed from a
glass disc. To prepare them between 2.7 to 3 g of sample powder was put in a crucible oven at (1050ºC) for one night, and
then weighted to obtain the Loss of Ignition (LOI) value. This calcinated samples were then used for preparing the fused



lithium tetraborate ($Li_2B_4O_7$) glass disc. To prepare them, we need to weight 6.0000g ± 0.0005g of lithium tetraborate and 1.2000g ± 0.0005 of calcinated sample. Both were put in an agate mortar and pound for 3 minutes to obtain a homogenised

powder. The powder was poured in a platinum crucible and in the automated glass bead-casting machine at University of Geneva (Pearl-X'3) to obtain a glass disc.

The trace elements were analysed from a pressed disc, assembled by weighting 3.000g ± 0.0005 g of wax and 12.000g ± 0.0005g of non-calcinated sample powder. The mixture was poured in a closed plastic container and shaken for 3 minutes to obtain a homogenised powder. The powder was then poured in a hydraulic press and 1 tonne of pressure was applied on it

during approximately 20 seconds, performed at University of Geneva. Accuracy of the analysed discs is 0.4 wt.% for the major elements and 1 to 3 ppm for the trace elements, assessed by analyses of standard reference materials.

### 3.4 Rock-Eval pyrolysis

The quality and quantity of preserved organic matter (OM) were determined in 237 bulk rock powders using the equipment Rock-Eval 6 at ISTE – UNIL, following the method described by Behar et al. (2001) and using the IFP 160000 standard.

Aliquots of samples were placed in an oven and first heated at 300ºC under an inert atmosphere, and then gradually pyrolyzed up to 650 ºC. After the pyrolysis was complete, the samples were transferred into another oven and gradually heated up to 850ºC in the presence of air, analysing the $CO_2$ and hydrocarbon (HC) concentration during all the process. The calculated parameters included total organic carbon content (TOC, wt.%), hydrogen index (HI, mg HC $g^{-1}$ TOC), oxygen index (OI, mg $CO_2$ $g^{-1}$ TOC), and $T_{max}$ (ºC) according to Espitalié et al. (1985) and Behar et al. (2001).

### 3.5 Carbonate carbon and oxygen stable isotopes

Carbonate carbon and oxygen stable isotope analysis ($\delta^{13}C_{carb}$ and $\delta^{18}O_{carb}$) of whole rock powders containing > 10 wt.% $CaCO_3$ ($n = 237$) were performed at the stable isotope laboratories of the Institute of Earth Surface Dynamics of the University of Lausanne (IDYST-UNIL). The used equipment was a Thermo Fisher Scientific Gas Bench II carbonate preparation device connected to a Delta Plus XL isotope ratio mass spectrometer. The $CO_2$ extraction was done by reaction with phosphoric acid

at 70°C. The stable carbon and oxygen isotope ratios were reported in the delta ($\delta$) notation as the per mil (‰) relative to the Vienna Pee Dee belemnite standard (VPDB), where $\delta = (R_{sample} - R_{standard})/R_{standard} \times 1000$ and R = $^{13}C/^{12}C$ or $^{18}O/^{16}O$. The $\delta^{13}C_{carb}$ and $\delta^{18}O_{carb}$ values were standardized relative to the international VPDB scale by calibration of the reference gases and working standards with international reference materials NBS 18 (carbonatite, $\delta^{13}C$ = –5.04 ‰, $\delta^{18}O$ = –23.00 ‰) and NBS 19 (limestone, $\delta^{13}C$ = +1.95 ‰, $\delta^{18}O$ = –2.19 ‰). Analytical uncertainty (1 sigma), monitored by replicate analyses of

the international calcite standard NBS 19 and the laboratory standard Carrara Marble ($\delta^{13}C$ = +2.05 ‰, $\delta^{18}O$ = –1.7 ‰), was not greater than ±0.05‰ for $\delta^{13}C$ and ±0.1‰ for $\delta^{18}O$.



### 3.6 Organic Carbon stable isotopes

The organic carbon stable isotope ratios ($\delta^{13}C_{org}$ values in ‰ vs. VPDB) were determined in 155 samples, which were previously decarbonated by treatment with 10% v/v HCl, thoroughly washed with deionized water and dried (40 °C, 48 h).

The $\delta^{13}C_{org}$ measurements were performed at the IDYST-UNIL by elemental analysis/isotope ratio mass spectrometry, using a Carlo Erba 1108 (Fisons Instruments, Milan, Italy) elemental analyzer connected to a Delta V Plus isotope ratio mass spectrometer via a ConFlo III split interface (both of Thermo Fisher Scientific, Bremen, Germany) operated under continuous helium flow (Spangenberg and Herlec, 2006). The calibration and normalization of the measured $\delta^{13}C$ to the VPDB scale was performed with international reference materials and UNIL in-house standards (Spangenberg and Herlec, 2006; Spangenberg,

2016). The repeatability and intermediate precision were better than 0.1 ‰ for $\delta^{13}C_{org}$.

### 4 Results

### 4.1 Stratigraphy and sedimentology

Belsué (BS) stratigraphic succession records the interfingering between prodelta (Arguís Fm.) and deltaic sediments (Belsué Fm.). The Arguís Fm. are highly bioturbated marls and silts, often rich in glauconite, with sparse bioclasts (*e.g.* bivalves), and

oxidized organic matter (OM) fragments. Sandstone beds (Belsué Fm.) are interlayered within the marls, forming two major coarsening and thickening upwards sequences that consist of medium sandstone beds (5-10 m thick), with sharp erosion base, parallel stratification, undifferentiated ripples, and glauconite rich horizons (Fig. 4). The Arguís marls are interpreted as prodelta deposits in a poorly circulated and relatively deep marine environment (Millán et al., 1994). The marls prelude deltaic mouth bars (Belsué Fm.) where the fluvial component predominates, although local effects from storms and tides are observed

(Millán et al., 1994; Castelltort et al., 2003). Calculated paleocurrents show a corrected east/south-east sediment supply source, in agreement with previous studies (Puigdefàbregas, 1975; Lafont, 1994; Millán et al., 1994; Garcés et al., 2014). Both formations are interpreted as a mixed delta-carbonate ramp system prograding westward spanning from Bartonian to Priabonian (Castelltort et al., 2003).



Figure 4: Stratigraphic logs of Belsué – East (BS-E), Belsué – West (BS-O), and Yebra de Basa, with a more complete high-resolution log (Yebra de Basa – HR). Red lines represent flooding surfaces (FS) and green lines the maximum flooding surfaces (MFS) correlation. Facies interpretation of the sedimentary logs are represented by grey bars, being more proximal the grey bar and more distal the white. Abbreviations used: Magneto. for magnetostratigraphy, Lithostrat. for lithostratigraphy; and Seq. Strat. for Sequence stratigraphy.

On the northern margin of the basin, the Yebra de Basa (YB) section starts with laminated blue marls (Larrés Fm.) that are interlayered by sparse siltstone beds and two dark levels rich in organic matter (56-60 m in Yebra-HR). The Larrés Fm. transitions to the Sabiñánigo sandstone (SS), that is composed by two thickening and coarsening upwards sequences. The sandstone beds present planar and through cross-stratification, erosion scours, as well as sigmoidal beds and flasser-wavy stratification. The upper boundary (*ca.* 205 m in Yebra HR) is marked by a sharp contrast towards a highly bioturbated and fossiliferous horizon (hard-ground), leading to the deposition of laminated grey-blue marls (Pamplona Fm.), less interlayered with siltstones beds, but richer in fossiliferous horizons (Turritella *sp.* mainly). The system is interpreted as a deltaic siliciclastic





shelf prograding W-SW (Roigé et al., 2018), defined as a fluvial delta with local tidal rework. The deltaic system ended abruptly with a major flooding that that formed a hard-ground and led to the Pamplona prodelatic marls deposition (Lafont, 1994; Puigdefàbregas, 1975).

Facies associations were described by combining the observations in the field and the available information by several previous
studies in the Jaca basin (Millán et al., 1994; 2000; Castelltort et al., 2003; Lafont, 1994; Puigdefàbregas, 1975; Boya, 2018). Using the vertical variations of facies in our sections we defined the depositional sequences that record the cycles of the shoreline's progradation/retrogradation (P-R). Here, we used the smallest correlatable sequences, which are termed parasequences when bounded by the two shallowest facies (flooding surface, FS, van Wagoner, et al., 1988, 1990), or genetic units when bounded by the two deepest facies (maximum flooding surface, MFS, Homewood, et al., 1992). The sequence
stratigraphic interpretation is summarized in Fig 4, where parasequences thickness from Belsué and Yebra de Basa vary from a few to tens of meters (5 to 50 m), and its stacking pattern defines two main P-R cycles in both sections.

**4.2 Clay Minerals**

In both sections, Belsué and Yebra, more than 90% of the total recorded clay mineral assemblage correspond to the sum of Chlorite, Chlorite/Smectite (CS), Mica, or Illite/Smectite (IS) (Fig. 5). This association of clay minerals is characteristic of
dominant physical erosion (Adatte et al., 2000). Mica is the most common clay in both sections (40% to 65%), followed by Chlorite (10 to 36%). In contrast, the percentage of Kaolinite is very low (<5%). In Belsué, both progradations show different Chlorite concentrations, being higher in the upper part. At Yebra, we observe an increase in Mica that coincides with the OM peak. The absence of smectite indicates it has been transformed during diagenesis into CS or IS mixed layers, and its percentage, 20-30% in our section, can be used as a burial estimation (Kübler, 2000). Kaolinite is positively correlated with
the deltaic progradation, indicating that kaolinite could be predominantly transported.





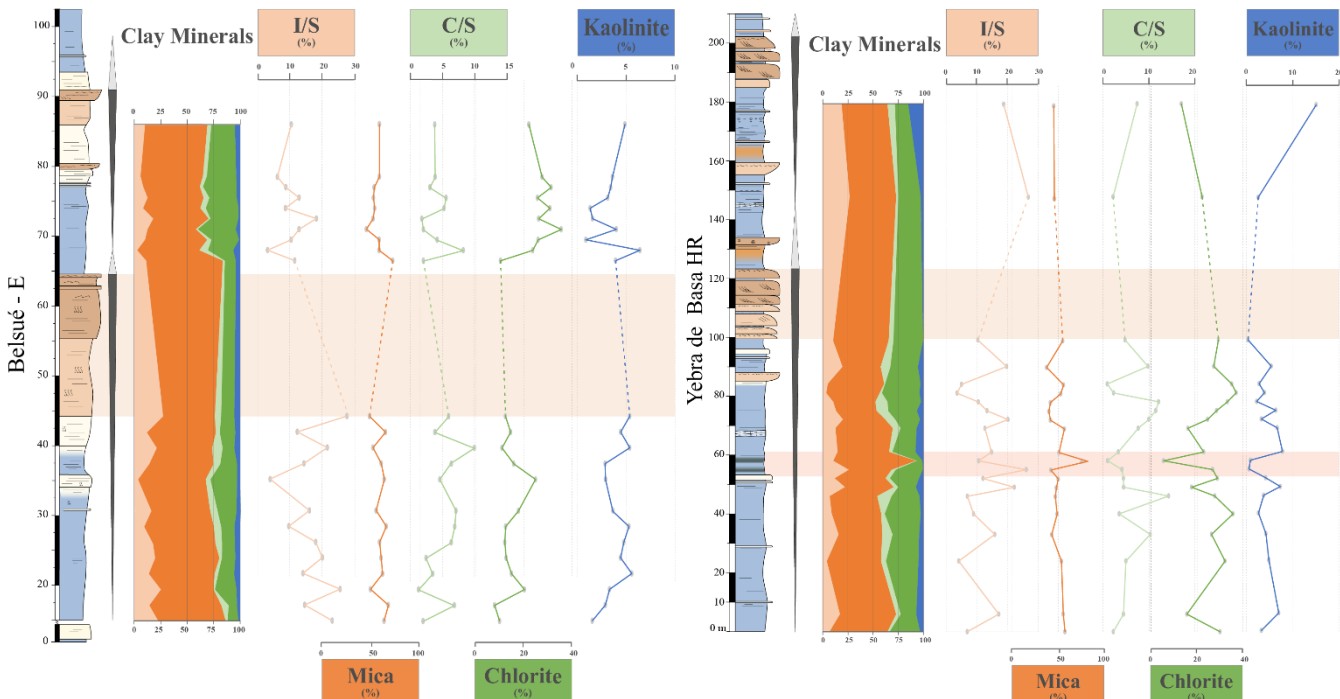

**Figure 5: Schematic stratigraphic log of Belsué-E (left) and Yebra de Basa – HR (right) with clay mineral assemblages C/S (Chlorite/Smectite), Chlorite, I/S (Illite/Smectite), Mica or Illite and Kaolinite. Highlighted in pale red the organic rich interval in Yebra de Basa, and in pale brown the main sandstone levels. The progradation-retrogradation cycles (P-R) are drawn with grey and white triangles.**

**4.3 Major and trace elements**

The major and trace elements have been normalized to aluminium (Al) to limit the dilution effect caused by different proportions of terrigenous sediment components (van der Weijden et al., 2002; Fig. 6). At BS, two increasing pulses of detrital major (Si, Fe, K, Ti) and trace elements (Mn, Sr) are concomitant with both deltaic progradations (Fig 6). The similar trend of Calcium (Ca) suggests an extrabasinal origin, likely from the eroded Mesozoic or Palaeocene carbonate platforms. Only the potassium (K) show a negative trend compared to the detrital elements, likely related to clay abundance. At YB, the high TOC interval (depicted in red in Figure 6) coincides with a relative decrease of the major Si, Ca, Ti, and K, and the trace Sr, Zr and Sn, normalized to aluminium. In contrast, the OM-related elements increase, such as the V/Cr ratio, which is related with organic matter and suboxic/anoxic conditions (van der Weijden et al., 2006), or the Ni/Co ratio, which is related to biogenic production (Tribovillard et al., 2006).




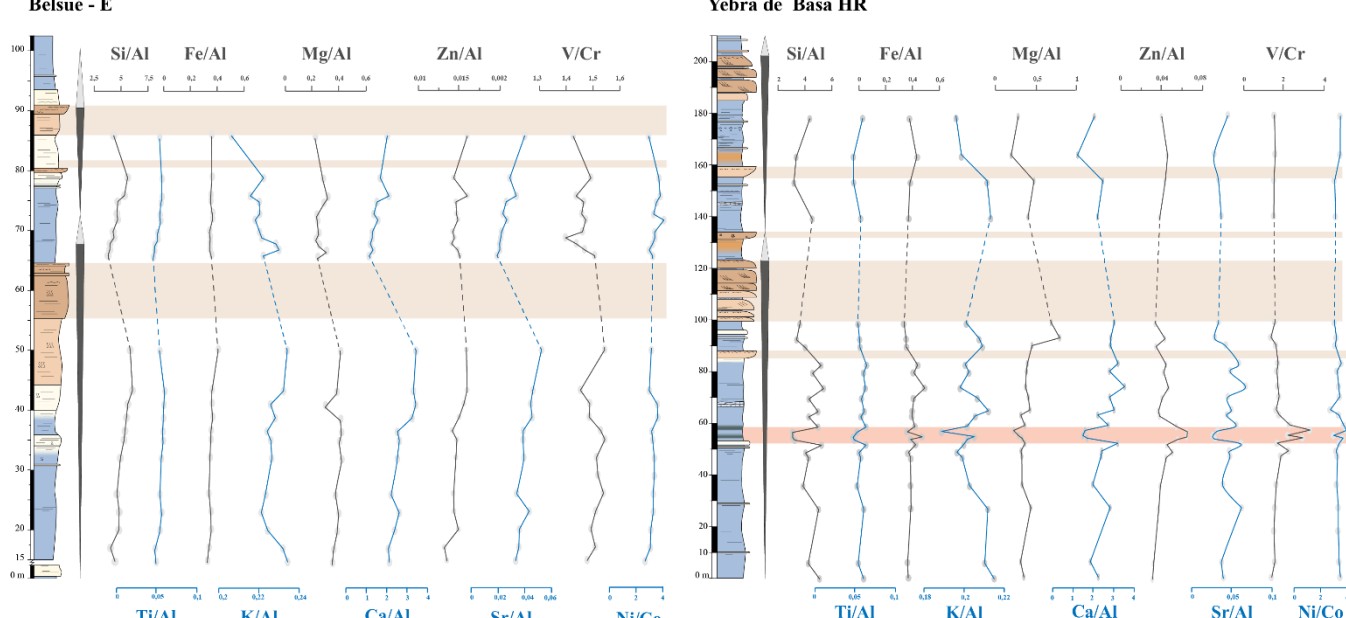

**Figure 6: Stratigraphic profiles of Belsué - E and Yebra de Basa HR with the normalized major element concentration (Si, Fe, Mg, Ti, K), trace elements (Sr, Zn), and trace element ratios (Ni/Co, V/Cr and U/Th). Highlighted in pale brown de sandstone levels, and in red the organic matter rich interval in Yebra de Basa. The progradation-retrogradation cycles (P-R) are drawn with grey and white triangles.**

## 4.4 Geochemistry

### 4.4.1 Organic matter content, type, and evaluation

The overall total organic carbon content (TOC) is low in both sections (average <0,5 wt. %; Fig. 7). At BS, the TOC values range from 0.06 to 0.38 % (average $0.2 \pm 0.07$ wt. %). The lower one shows a decreasing trend of TOC that follows the deltaic progradation, showing that with increasing clastic material the OM concentration decrease. The upper section also depicts this trend, and higher and more stable TOC values (~0,3 wt.%) that are associated with marly prodelta deposits. In the other hand, YB TOC values range from 0.14% to 1.3% (average $0.3 \pm 0.13$ wt. %). The most prominent feature is a dark-grey marl interval (Fig. 7) with a organic matter peak that reaches values higher than 1 wt.%, and is associated with a negative isotopic excursion (Fig. 7). Apart from this major excursion, values keep close to 0.3% and no other major variations occur along the section. In the high-resolution part of the section, there are small oscillations varying up to ±0.1 wt.%.





**Figure 7: Stratigraphic logs of Belsué and Yebra de Basa including the results of total organic carbon (TOC wt.%), stable carbonate isotopes (δ $^{18}$O$_{carb}$ and δ $^{13}$C$_{carb}$), and organic carbon stable isotopes (δ $^{13}$C$_{org}$). The two progradation-retrogradation cycles referred in the text are drawn next to the stratigraphy, note the change in the scale. Highlighted in grey the isotopic signature and in pale red are the OM-rich interval coeval to the MECO thermal peak. The wide coloured lines correspond to the 3-point moving average, whilst the central part of the Yebra de Basa section due to its high resolution has a 7-point moving average curve. Magnetostratigraphic logs from Vinyoles et al., (2021) and Garcés et al., (2015).**

The values of T$_{max}$ classify the HI (Hydrogen Index)/OI (Oxygen Index) in terms of OM origin and quality (Espitalié et al., 1985). Our results show that samples range from 422ºC to 445ºC (Fig. 8), with some exceptions reaching almost 460ºC. This indicates that the character of the preserved OM is generally immature or within the oil zone (Fig. 8). The hydrogen index (HI) in YB and BS is generally below 100 (average 65 mg HC/g TOC), which falls in Type III and Type IV zone of organic matter origin, characteristic of terrestrial plants (Espitalié et al., 1985, Fig. 8). Some samples in Belsué (BS-W) record higher HI than 150, this is probably related with a slightly different depositional condition between the sections (more distal in the W). The oxygen index (OI) shows a similar trend than HI values, keeping values below 100 (average 82 mg CO$_2$/g TOC), but more dispersed than HI values. Altogether points out towards a recycled source and/or terrestrial origin of OM, for both YB and BS sections.



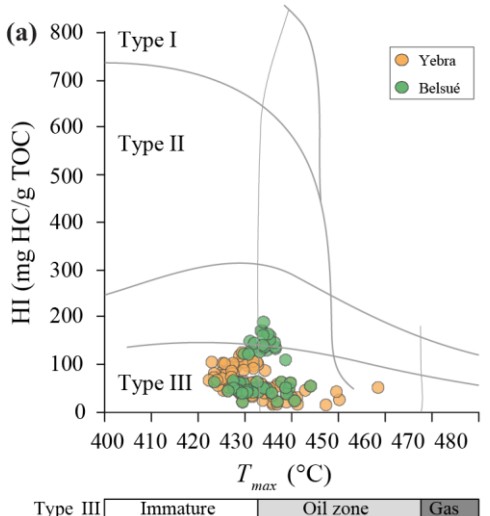 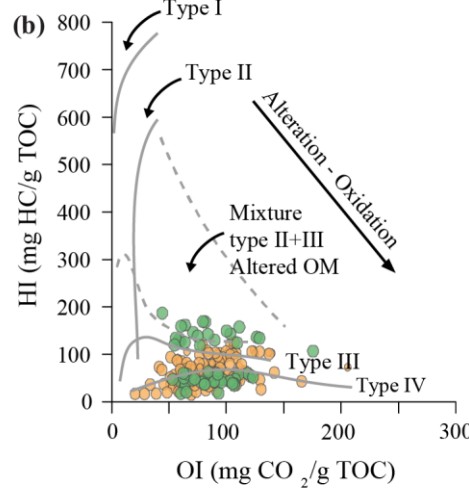

**Figure 8: (A) The scatter plot of hydrogen index (HI) vs. $T_{max}$ values allows to discriminate the different kerogen types, (B) whereas the scatter plot of hydrogen (HI) vs oxygen index (OI) allows to assess the OM origin and maturity. Here we display the kerogen types (A) and the OM origin (B) from Yebra de Basa (orange) and Belsué (green). Samples with lower TOC than 0,2 % have been excluded. Reference lines for kerogen maturity based on Espitalié (1986).**

### 4.4.2 Carbonate carbon and oxygen stable isotopes

At BS, the $\delta^{13}C_{carb}$ values range from -2.6 to -0.2‰ (-1.5 ± 0.4 ‰), and the $\delta^{18}O_{carb}$ values range from -5.7 to -2.9‰ (average -4.4 ± 0.6 ‰; Fig. 9). Oxygen isotope ratios show a gradual decreasing trend during the first two deltaic progradations (-1.5 ‰), coinciding with a very gradual and small positive trend of the $\delta^{13}C_{carb}$. After the second progradation, $\delta^{18}O_{carb}$ rapidly returns to more positive values, maintained until the top of the section. This steadiness is not followed by the $\delta^{13}C_{carb}$ results that record a pronounced positive shift between 125 and 150 m height (+1 ‰). The $\delta^{18}O_{carb}$ values show a gradual decreasing trend during the first two deltaic progradations (of -1.5 ‰), coinciding with a very gradual and small positive trend of the $\delta^{13}C_{carb}$ values. After the second progradation, $\delta^{18}O_{carb}$ rapidly returns to more positive values, maintained until the top of the section. This steadiness is not followed by the $\delta^{13}C_{carb}$ results that record a pronounced positive shift between 125 and 150 m height (of +1‰). At YB, the $\delta^{18}O_{carb}$ values range from -6.4o to -4.2‰, (average -4.9 ± 0.4‰), and the $\delta^{13}C_{carb}$ values from -1.8 to 0.6‰ (-0.8 ± 0.4‰) (Fig. 9). Small oscillations (±0.5 ‰) of the $\delta^{18}O_{carb}$ dominate the lower part of the section, and ends with a significant negative shift at 285 m. There, the $\delta^{13}C_{carb}$ values decrease by 0.8 ‰ and the $\delta^{18}O_{carb}$ values decrease by 1.3 ‰. This level is organic rich (1.0–1.5 wt.% TOC) and shows also decrease of 2‰ in the $\delta^{13}C_{org}$ values (see below). Above the organic-rich interval, the $\delta^{13}C_{carb}$ and $\delta^{18}O_{carb}$ values return to pre-event background values, with a shift to higher values towards the base of the Sabiñánigo sandstone, where the $\delta^{13}C_{carb}$ values reach the maximum value of 0.6‰. A gradual decrease of the $\delta^{18}O_{carb}$ values coincide with the recurrent progradation events evidenced by the Sabiñánigo sandstone. In contrast, the $\delta^{13}C_{carb}$ follows a stable trend around -1‰ until the top of the section. Above the Sabiñánigo sandstone and within the Pamplona marls, the $\delta^{18}O_{carb}$ values show no variance until the top of the section





### 4.4.3 Organic carbon isotopes

At BS section, the $\delta^{13}C_{org}$ values range from -26.3‰ to -22.6‰ (-24.6 ± 0.5‰; Fig 6), including two groups of $\delta^{13}C_{org}$ values. In the first group, formed by the samples before the first siliciclastic progradation (0–50 m), the TOC content is low (0.1–0.3 wt.%) and the $\delta^{13}C_{org}$ values relatively higher (-25 to -24‰). The second group is formed by samples between the deltaic

propagations (65–85 m), which have higher TOC content (0.3–0.5 wt.%) and lower $\delta^{13}C_{org}$ values (~ -26‰). At YB, the $\delta^{13}C_{org}$ vary between -27.2 and -23.7‰ (-24.9 ± 0.8‰; Fig 6), whose lowest value of -27.2‰ was measured within the OM rich level at 285 m. The $\delta^{13}C_{org}$ values increase upwards till the base of the Sabiñánigo sandstone, where they show a negative spike, coinciding with the positive shift of the $\delta^{13}C_{carb}$ and $\delta^{18}O_{carb}$ values. Then the $\delta^{13}C_{org}$ values first return to the pre-event background values, and then show a negative excursion of to 2‰ in the last two samples of the Sabiñánigo sandstone.

## 5 Discussion

### 5.1 Primary versus diagenetic signals

Chemostratigraphy arises a multitude of possible influences from differences in biological, diagenetic, and physico-chemical factors that can mask the primary signal (Wendler, 2013). To better discern primary versus altered signals, it is necessary to understand the factors controlling the primary isotopic composition and assess the potential extent of diagenetic overimprint.

Oxygen isotopes in carbonates are controlled by the temperature of formation, the $\delta^{18}O$ value of the carbonate-precipitating fluid ($\delta^{18}O_w$), the mineralogy (*e.g.,* higher $\delta^{18}O_{dolomite}$), and any kinetic effect manifested during the precipitation (*e.g.,* pH, salinity; Swart 2015). $\delta^{18}O$ is generally used as a temperature proxy in the marine realm, even though it is more prone to alteration (Fio *et al*. 2010). In contrast, carbon isotopes are not thought to be directly influenced by temperature and are generally more resistant to diagenetic processes (Schrag *et al*., 1995; Swart, 2015). However, $\delta^{13}C$ values are also controlled

by kinetic effects, mineralogy, and mostly by the $\delta^{13}C$ trace from the dissolved organic carbon (DIC; Wendler, 2013). The isotopic signal from the DIC indicates the source of this carbon, especially the type of oxidized/respired organic matter components (Swart, 2015). In proximal depositional environments, however, this could be modified by (1) organic matter productivity and burial rate, (2) extrabasinal carbonate input, (3) water circulation/stratification and evaporation, (4) terrestrial runoff and weathering (Saltzman et al., 2012, Läuchli et al., 2021). Considering this, $\delta^{13}C$ is usually used as a global correlation

tool since it can register eustatic sea-level fluctuations, changes in weathering flux, or significant perturbations in the global carbon cycle (*e.g.,* volcanic $CO_2$ input; Wendler 2013 and references therein).



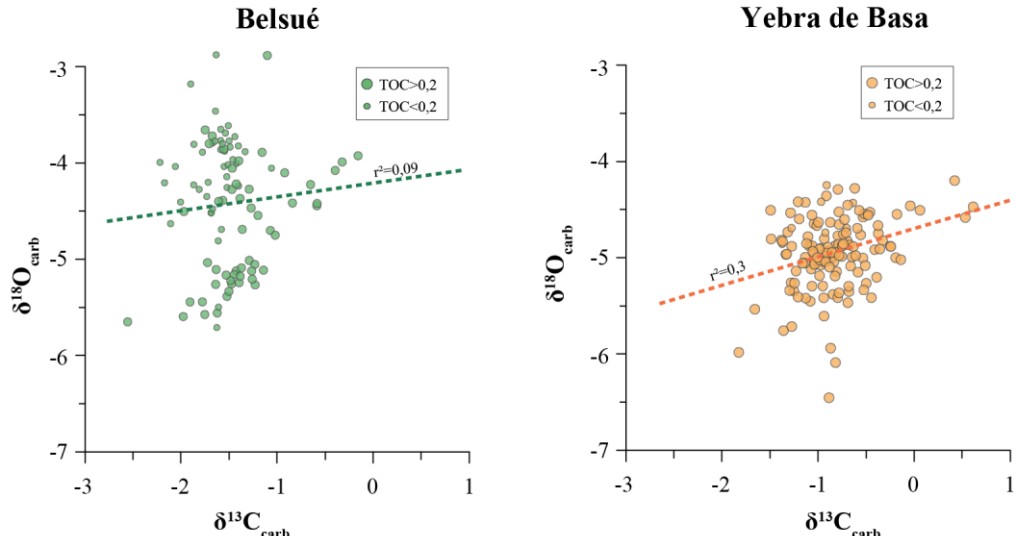

**Figure 9: $\delta^{18}O_{carb}$ - $\delta^{13}C_{carb}$ scatterplot of all Belsué and Yebra de Basa samples. The regression lines are given for reference with the square correlation coefficients ($r^2$). The size of the symbols is small for samples with TOC < 0.2 wt.% and big for samples with TOC > 0.2 wt.%.**

During carbonate diagenesis, the cementation, dissolution and re-precipitation (neo-formation), due to the interaction with post-depositional fluids —probably depleted in $\delta^{18}O$ or at higher temperature, or both— can alter the primary oxygen isotope composition (Schrag et al., 1995; Marshall, 1992). The carbon isotope composition can be also modified by differences in the contribution of biomass (marine *vs.* terrestrial), of carbonates (*e.g.* allochthonous clasts), and/or $^{13}$C-depleted DIC derived from oxidation/respiration of organic matter (Marshall, 1992; Schrag et al., 1995; Wendler, 2013).

One method to assess the degree of diagenetic alteration is to evaluate the correlation between $\delta^{13}C$ and $\delta^{18}O$ values (Brasier et al., 1996). Statistically, a non-significant positive correlation (r > 0.6) indicates that a diagenetic overprint of the primary isotopic signature can be excluded (e.g. Fio et al. 2010). Our values from Belsué and Yebra de Basa show no significant statistical $\delta^{13}C$-$\delta^{18}O$ correlation ($r^2$<0.3) for both sections, probably suggesting non or reduced diagenetic modification of the primary signals (Fig. 9). Additionally, as proposed by Kubler and Jaboyedoff (2000), the illite crystallinity serve to assess the

degree of possible alteration of the mineral assemblage by estimating the stage of diagenesis that has reached the sample. These authors compared clay mineral assemblages, illite crystallinity, and OM parameters to define four diagenetic zones. The presence of smectite within the illite-smectite (IS) mixed layers in our samples is between 20 and 30%, which according to the Kübler and Jubeyedoff (2000) zonations, fall within the 3$^{rd}$ diagenetic zone, i.e. shallow diagenesis (*ca.* 60 to 80ºC). Another diagenetic indicator is the maximum temperature ($T_{max}$) reached during the Rock-Eval Pyrolysis (S2), which marks the

maturity of the organic matter. We checked $T_{max}$ using only the samples with relatively high organic matter content (TOC>0.5wt.%; S2>0.2). The measured $T_{max}$ values are 440ºC, corresponding to the beginning of the oil window (*ca.* 60ºC; Espitalié et al., 1985; see Fig. 8), which also agree with vitrinite reflectance and Raman measurements from this area (Labaume et al., 2016).





All the diagenetic evidences, among them the isotopic values correlation, the illite crystallinity, and the $T_{max}$, suggest that
diagenetic overprint is small. Therefore, the primary isotopic signal is preserved in both sections and the geochemical results
can be safely used as proxies to study paleoenvironmental conditions, and eventually be compared to global key isotopic curves
during the MECO event.

## 5.2 MECO isotopic record

In Belsué, the oxygen isotopic record shows a general trend towards more negative values from base to the middle sandstone
units. This trend is intensified by a negative $\delta^{18}O_{carb}$ shift of ~1 ‰ prior to the sandstone unit, just in the chron transition C19r-
C18n.2n (Fig. 10). In Yebra, the oxygen isotopic record shows a small-scale variability, consistent with local effects, but then
a negative $\delta^{18}O_{carb}$ shift of ~1.2 ‰ occurs close of the deltaic progradation (Fig. 10). Either in Belsué or Yebra de Basa the
MECO zenith (around the magnetic reversal C19r-C18n.2n; Bohaty et al., 2009; Edgar et al., 2010; Henehan et al., 2020) is
represented by the progradation of the deltaic facies over the prodelta, i.e. the deltaic facies of Belsué Fm. in Belsué and the
Sabiñánigo sst. in Yebra (Fig. 10), and no isotopic data was taken in this interval. Nevertheless, the onset of the main thermal
event, just before the sandstone occurrence is preserved as the negative excursion. After the sandstone progradation both
Belsué and Yebra oxygen values recover and become similar than before the excursion (Fig. 10).





**Figure 10: Oxygen isotopic (δ¹⁸O$_{carb}$ ) correlation panel for the studied sections (Belsué and Yebra de Basa) with MECO target curves from Alano (Italy, Tethys ocean, Spofforth et al., 2010), ODPS 1051 (N Atlantic Ocean; Edgar et al., 2010), ODPS 702 (S Atlantic Ocean; Bohaty et al., 2009) and ODPS 738 (S Indic Ocean; Bohaty et al., 2009). Data from the bulk and fine sediments fractions. Highlighted in red the organic matter (OM peak) rich interval in Yebra de Basa. The two progradation-retrogradation cycles referred in the text are drawn with grey and white triangles.**

The trend observed in the SPFB is shared with most of high-resolution offshore and nearshore isotopic records (Bohaty et al., 2009; Bohaty and Zachos, 2003; Edgar et al., 2010, 2020; Spofforth et al., 2010; Jovane et al., 2007; Giorgini et al., 2019; Galazzo et al., 2014) that also show the same trend towards more negative δ ¹⁸O values intensifying during the MECO peak (Fig. 10). Similarly, the end of the event is defined by a rapid increase in δ ¹⁸O$_{carb}$ values of ∼1 ‰, similar to other sites worldwide (Bohaty et al., 2009; Galazzo et al., 2014; Edgar et al., 2010, 2020; Giorgioni et al., 2019; Spofforth et al., 2010). Contrarily to the agreement between our new data and most of available oxygen isotopic records, δ¹³C$_{carb}$ results do not show a clear correlation with global target curves (Fig. 7). On one hand, Belsué section records a positive δ¹³C$_{carb}$ excursion, presenting a delay respect the δ¹⁸O$_{carb}$ minimum, and Yebra show a prominent positive δ¹³C$_{carb}$ excursion just before the main deltaic progradation (320-340 m from YB). On the other hand, most of the oceanic geochemical records show a small Carbon Isotopic Excursion (CIE) at the MECO peak of warming (∼40 Myr; Westerhold and Röhl, 2013; Bohaty et al., 2009; Spofforth et al., 2010), but before and after the carbon is highly variable, showing opposite trends between hemispheres (Henehan et al.,



2020; Giorgini et al., 2019). This CIE, like in other sites in the northern hemisphere (Sppofforth et al., 2010; Giorgini et al.,
2019), is not well represented in our data. Therefore, our $\delta^{13}C_{carb}$ results seem to confirm the fact that the MECO is not associated with a large input of depleted $^{13}C$ carbon in the environment as previous studies suggest (Henehan et al., 2020), although an alternative is the CIE could be masked by the sandstone progradation itself.

Instead of a global origin, however, these variations could be related with local DIC sources or changes in the mineralogy (*e.g.*, dolomite). Here, given the proximity of the continent in the shelf environment, fresh water riverine (and groundwater) input
fluctuations could have altered the carbon isotopic composition of the DIC (Saltzman et al., 2012; Wendler, 2013 Läuchli et al., 2021), representing the terrestrial contribution to the oceanic carbon reservoir. Thus, given the observed $\delta^{13}C_{carb}$ variations and the peculiar carbon isotopic record during the MECO, we focus our correlation on the $\delta^{18}O_{carb}$ record.

### 5.3 The organic matter peak in Yebra de Basa

In Yebra de Basa (YB) section, an increase in organic matter content at the 280-290 m interval (up to 1.5 wt.% TOC) is
associated with a negative isotope excursion of -1.5‰ for $\delta^{18}O_{carb}$, -2.0‰ for $\delta^{13}C_{org}$ and -0.8‰ for $\delta^{13}C_{carb}$ values (Fig. 7 and 9). The OM rich interval occurs 50 meters below the main Sabiñánigo sandstone progradation in Yebra, and it is not coincident with the main increase of detrital input, marked by an increase in grain size. This boost in organic matter burial is also observed in the Neo-Tethys region, like in Italy (Spofforth et al., 2010) and the Crimea-Caucasus (Benyamovsky et al., 2012), which may had played an important role in carbon drawdown and rapid cooling after the MECO event (Bohaty et al., 2009, Henehan
et al., 2020).

Several possibilities could explain the presence of an OM-rich interval before a deltaic progradation. First, a significant freshwater input in a restricted basin can lead to water stratification where anoxic conditions are favoured, this resulting in an increase of OM preservation, independently of its source. Nevertheless, the slight increase in redox-sensitive elements (V and Mo, Fig. 6) is too limited to support the development of water stratification and the resulting suboxic-anoxic conditions
(Tribovillard et al., 2006). Second, the enhanced freshwater input could have increased nutrient availability and the consequent marine productivity. We, however, reject this hypothesis because our organic matter analyses show a clear terrestrial compound (low HI-OI) and no sign of nutrient availability increase (Ni concentration; Tribovillard et al., 2006). Our preferred explanation is that the OM peak could be related to a significant increase in detrital input and terrestrial OM. The presence of several dark-marl beds westwards suggests it was not a unique episode, but instead a series of recurrent events (Boya, 2018).
In addition, the terrestrial origin is also supported by the strong correlation (r>0,7) observed between the siliciclastic elements (Al, Ti, $Fe^{3+}$) and the TOC or all the OM-related trace elements (V, Mo, Ba, Th; Tribovillard et al., 2006). Despite this, the isotopic results do not agree with this correlation, because pre-Miocene marine OM had lower $\delta^{13}C$ than terrestrial OM (Popp et al., 1989). Thus, an alternative explanation for the negative $\delta^{13}C_{org}$ and $\delta^{13}C_{carb}$ excursion may be an increased input of organic matter released from soils containing bacterial biomass with low $\delta^{13}C_{org}$ values (Fio et al., 2010). This is in agreement
with our Rock evaluation and geochemical results, that point towards a terrestrial origin for this organic matter.



As a result, the Sabiñánigo sandstone represents a singular deltaic event embedded in long lasting prodelta conditions (Vinyoles et al., 2021) in which no evident organic events occur. Therefore, we interpret the occurrence of the OM rich level just before the Sabiñánigo sandstone as a first indicator of a shift towards a setting with more fluvial conditions, being the first evidence of the main MECO excursion in the region.

## 5.4 MECO response in the South Pyrenean Foreland Basin

The integration of available age constraints (Garcés et al., 2014; Vinyoles et al., 2021) and the new high-resolution isotopic record show that MECO's warming peak (~ 40 Ma) is associated with isochronous progradation, which can be followed all along the SPFB source-to-sink system (Fig. 11; Vinyoles et al., 2021). In the The Tremp-Graus basin the Escanilla fluvial system was fed by the Sis-Gurp and Pobla alluvial systems, where a grain size increase is recorded at *ca.* 40 Ma (Whittaker et al., 2011). Downstream, in the time-equivalent sections in the Ainsa basin, an anomalous amalgamated Olsón sheet stands out from the landscape as a continuous and thick conglomeratic bed, interpreted as a stacking of several braided river channels (Fig. 11; Verité, 2019; Labourdette et al., 2011; Puigdefàbregas, 1975; Vinyoles et al., 2021). In the deltaic counterparts (Jaca basin), a significant progradation of deltaic deposits on top of slope marls is observed in our studied sections (BS and YB; Lafont, 1994; Puigdefabregas et al., 1975, Vinyoles et al., 2021). Finally, in the deeper sink environments of the Jaca and Pamplona basins, the correlation with the turbiditic systems is still debated and needs further research.

Previous works (Puigdefàbregas, 1975; Lafont, 1994) interpreted these deltaic sequences as eustatic fluctuations of the relative sea level, which can relate to different possibilities, such as thermal expansion or glacioeustasy. Ephemeral ice sheets in Antarctica during the Middle Eocene are likely, and it seems plausible that the progressive shift towards icehouse conditions could have significant implications during the MECO (Edgar et al., 2007; Huyghe et al., 2012; Baatsen et al., 2020). However, considering the temperature increase interpreted during the MECO zenith (+4 to 6ºC; Bohaty et al., 2009), we should expect a sea-level rise (ice caps melting and thermal expansion) instead of the observed regression and system progradation.

Alternatively, an abrupt increase in sediment supply can also explain a progradation of deltaic systems. Several studies observed that the main Paleogene hyperthermals are often associated with an enhanced flux of terrigenous material interpreted as a boost of the hydrological cycle and higher seasonality (Schmitz et al., 2001; Chen et al., 2018; Foreman et al., 2017; Pujalte et al., 2015). Although the MECO is not an abrupt event like other hyperthermals, but instead a more extended period of gradual warming (*ca.* 500 kyr; Bohaty et al., 2009), we also observe this progradation focused during the warming peak (*ca.* 40 Ma). Accordingly, an explanation for the progradation is that the MECO prolonged warming produced an enhanced hydrological cycle that favoured sediment production and transport, thus leading to an increase in sediment supply and favouring the system progradation at the peak of the event. The nature of a greater sediment provision (Qs) should be originated upstream, for instance, linked to enhanced sediment remobilization (e.g. floodplain) or accelerated hillslope processes (Foreman et al., 2012).





**Figure 11: Correlation panel between Belsué, Yebra de Basa and Olsón section with the GPTS 2016 (Ogg et al., 2016). The stratigraphic sections are modified from Garcés et al. (2014) and Vinyoles et al. (2020). The oxygen isotopic record ($\delta^{18}O_{carb}$) from ODPS 738 and the MECO age constraints defined by yellow and blue bars are modified from Bohaty et al. (2009). The oxygen isotopic record ($\delta^{18}O_{carb}$) from Belsué and Yebra de Basa correspond to our results.**

Therefore, the coincidence in time of a basin-wide progradation in the SPFB and the MECO might implicate a link between them. Our geochemistry analyses also suggest a terrestrial origin for this OM, which point towards an increase in soil remobilization, erosion and transport in continental environments during the MECO event.





## 5.5 Global implications and correlation

The global impact of the MECO event in continental settings remains currently poorly documented, with only a few studies in continental environments performed around the globe (e.g., Bosboom et al., 2014; Mulch et al., 2015). In the North American plateau, a boost of precipitation during the MECO is derived from lower $\delta^{18}O_{carb}$ values (Mulch et al., 2015). In contrast, in the Tarim basin (China) a shift towards arid conditions has been interpreted from a reduction in fern palynomorphs (Bosboom et al., 2014). This aridification trend in central Asia differs from the documented Neo-Tethys ocean dynamic, where marine records show an increase in organic matter (OM) burial during the MECO peak and part of the post-MECO recovery (Spofforth et al., 2010; Giorgioni et al., 2019 Benyamovskiy et al., 2012). Increased sediment supply due to enhanced erosion and transport clearly provides a mechanism for more efficient burial of OM during this and other hyperthermals (Galy et al., 2007). If this enhanced OM burial is global or sufficiently widespread (it is absent in several sections, including Belsué in this study), it could represent an important mechanism to explain the carbonate $\delta^{13}C$ increase that is recorded globally during the post-event recovery and the associated rapid return to pre-event conditions, maybe playing an essential role in the drawdown of atmospheric carbon (e.g., Bohaty et al., 2009; Henehan et al., 2020; Sluijs et al., 2013; Edgar et al., 2020; Giorgioni et al., 2019; Spofforth et al., 2010).

Considering the long duration of the MECO event (*ca.* 500 kyr; Bohaty et al., 2009), some of the most important effects in the ocean occur during its peak phase, *e.g.* ocean acidification (Bohaty et al., 2009; Henehan et al., 2020; Arimoto et al., 2020) or OM burial (Giorgioni et al., 2019; Spofforth et al., 2010). In the SPFB, the continental progradation also occurred at the end of the event, supported by the sedimentological and geochemical evidences that show an increase of sediments delivered to the sea, including large amounts of organic matter of terrestrial origin. Hence, our work suggests a link between enhanced hydrological cycles and enhanced OM transport and burial, which possibly account for the observations of enhanced OM burial around the Neo-Tethys region. This response in sediment delivery rate, OM burial in shallow and restricted basins, as well as ocean acidification, has been previously documented for other early Eocene hyperthermals (Chen et al., 2018; Foreman et al. 2012, 2014; Pujalte et al. 2015; Foreman and Straub 2017; Honegger et al. 2020). Hence, the MECO, despite its important differences with the early Eocene hyperthermals, yet shares several attributes with them around the warming peak. In summary, our results point to a more intense hydrological cycle perturbing rainfall patterns in the Pyrenean region during the MECO peak, and leading to increased sediment supply, expressed by a major progradation of sedimentary systems and eventually, an increase in OM burial in the nearby oceanic basins.

## 6. Conclusions

In the South-Pyrenean Foreland Basin, an important progradation affected the entire sediment routing system from fluvial to deltaic environments at times of the Middle Eocene Climatic Optimum MECO. Here we present a new high-resolution multiproxy dataset, including stable isotopes, Rock-Eval, XRF, and clay minerals, covering the different MECO phases from two well dated key sections. The new stable isotopes records from Belsué (BS) and Yebra de Basa (YB) sections show a

significant negative shift in the shallow marine sediments, around the main warming peak of the MECO event, for the first time reported in the Pyrenean region. In Yebra de Basa, an organic-rich interval of terrestrial origin is found before the main deltaic progradation, and it is associated with a negative excursion in oxygen and carbon isotopes. The correlation of the

MECO, the basin-wide progradation, and our new geochemical results presents compelling evidence for a climatic driver, suggesting an enhanced hydrological cycle in the Pyrenean region that caused a boost in sediment and carbon export. This is in agreement with previous studies from the Neo-Tethys ocean that recorded an increase in organic matter burial during the peak of the MECO and early post-MECO.

Although the duration of the MECO and its isotopic signature differ with respect to early Eocene hyperthermals (e.g., PETM),

there are similarities around the warming peak that trigger a comparable response, including ocean acidification, OM burial or a boost in sediment supply export from land to sea. Nevertheless, further work is needed to understand the role of potential sediment supply increase from the proximal continental environments towards the deeper oceanic basins, and importantly, quantify sediment and organic export, and its relation with carbon burial and silicate weathering.

Our results support the view that high-accommodation settings in foreland basins are important recorders of

paleoenvironmental signals, even in shallow marine environments. Although certainly noisy, the fact that climate signals are preserved in these settings provide a range of potentially expanded sections that can be interesting complement to high-resolution but more condensed deep-sea paleoclimatic records. In particular, during high-$CO_2$ globally warm episodes of the Earth's history when the carbonate-rich oceanic records may undergo intervals of non-deposition or dissolution.

**Data availability**

All the data (stable isotopes, clay minerals, organic matter, major and trace elements) can be found in the supplementary material.

**Authors contribution**

SPC led fieldwork, sampling, sample preparation, data interpretation, and writing. LV contributed to the fieldwork, data interpretation, discussion, and writing. JES performed stable isotope analyses, data interpretation, and writing. TA performed

XRD analyses, data interpretation, and discussion. JV and AV contributed to the field work preparation, sampling and discussion. MT, SW and NS contributed to the discussion, and writing. CP contributed to fieldwork, discussion, and writing. AV and MG helped with magnetostratigraphic data interpretation and discussion. SC supervised the project, funding, interpretation and writing.



**Competing interest**

The authors declare that they have no competing interests.

**Acknowledgments**

The authors would like to acknowledge the *Societé de Physique et d'Histoire naturelle de Genève* and Equinor (grant to Castelltort) for financing part of the field missions. We also acknowledge Marta Roigé and Salvador Boya for their help during field campaigns and the long scientific discussions. We finally acknowledge Antoine de Haller from *Université de Genève*

for his help during XRF analyses.

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
