# Peer review of "Fluvio-deltaic record of increased sediment transport during the Middle Eocene Climatic Optimum (MECO), Southern Pyrenees, Spain"

_EGUsphere, 2022_

## Referee Comment (RC1)

**A review of:**
**Fluvio-deltaic record of increased sediment transport during the Middle Eocene Climatic Optimum (MECO), Southern Pyrenees, Spain**

Eric A. Barefoot

December 1, 2022

**Synopsis**

Using two new isotope and stratigraphic sections in the west-central Pyrenees, Cabré et al study the connection between the Middle Eocene Climatic Optimum (MECO) and sediment supply. Working in the Tremp-Jaca Basin, the authors collected geochemical and sedimentological data from two stratigraphic sections. These sections target two separate deltas, which were part of a common depositional system that deepened westward, created by subsidence in the Pyrenees foreland basin. The authors measured stable carbon and oxygen isotopes, trace and major elements, as well as organic matter composition and maturity. They synthesize these data into a new understanding of this basin during the MECO.

Their main findings are supported by two key observations:

1. The authors observe that the MECO coincides with two episodes of delta progradation in this basin. The authors assert that this is due to enhanced sediment supply due to changes in hydroclimate. They reject a suite of alternative hypotheses (enhanced uplifting, eustatic sea level, etc.)

2. The authors observe that patterns in stable isotopes of oxygen from their stratigraphic sections parallel patterns in the global ocean and other basins, patterns of carbon isotopes do not. Based on this observation, they assert that because the deltas are prograding into a restricted ocean basin, the carbon isotope signature is dominated by local effects (provenance, local chemistry) rather than a global input of depleted carbon. It is not made clear why oxygen isotopes are not affected by these processes. The authors assert that diagenesis effects have not altered their samples substantially.

Based on these main findings, the authors conclude that in the Iberian peninsula, climate-induced episodes of enhanced erosion are connected to transient warming and changes in hydroclimate. This bolsters similar findings from strata across the Pyrenees during the Eocene. The MECO is relatively long-lived ($\sim$ 500kyr) compared to its shorter counterpart hyperthermals during the EECO (e.g. PETM ($<$ 200kyr)). Given this, the authors suggest that the MECO may be a good test case for understanding feedbacks between the carbon cycle, hydrological cycle, and other Earth systems.

**Overall Comments**

I read this paper with interest, and a fair bit of initial skepticism. The authors have identified a good case study for an important problem. In many stratigraphic studies like this one, there are major issues of resolution and timing. This is particularly a problem with Eocene hyperthermals, where the hypothesized duration of a climate "episode" or "event" is brief enough ($< \sim$ 100kyr) that it can be distorted, disguised, or destroyed entirely by sediment transport processes in the sedimentary basin. On the other hand, if a climate episode is very long ($>$ 1 Myr), then there are few terrestrial basins that can record the full length of the event without major issues of changing subsidence rates and basin filling, etc. This basin, and episode of interest, appears to lie in a kind of optimal middle ground, where the MECO is long-lived enough that

one can be reasonably sure to capture the signal despite transport processes, and yet short enough given the basin size to ignore tectonic issues.

With the aid of this well-chosen field site, the authors have done a really nice job of constraining the problem. By leaning on a firm grounding in regional literature and collecting a comprehensive suite of data, the authors have persuaded me that their model is the most likely scenario. I found the observations to be appropriate, well-reasoned, and well-documented. The analysis of these observations is careful, and clearly articulated.

Overall, I think this is a very well-executed study, it will have substantial appeal to readers of Climate of the Past across disciplines. I expect it may spark some increased interest in studying the MECO, and may serve a touchstone for motivating studies that join the best insights of sedimentary geology and paleoclimate. I have a few minor comments and questions for the manuscript, but after those have been addressed, I think it would make a good contribution to CP.

The authors should feel free to reach out to me if they have any questions about my review.

**Minor Comments**

**comments on: Primary vs Diagenetic Signals & MECO isotopic record**

I appreciate the value of these two subsections of the discussion. It is important to make sure that your reader is aware that you have rigorously tested the alternative hypothesis. In this cased though, I think the order of these two subsections gets in the way of your message. Why have caveats dominate the first part of your discussion? Rather, I would suggest discussing your results (the carbon and oxygen isotopes), identify the discrepancy (the oxygen isotopes match global trends, the carbon do not), then discuss the possible mechanisms. I think by switching sections 5.2 and 5.1 will flow logically better, because it will allow you to place the discussion of your observations first, then follow up with evidence.

Additionally, I found the language in these two particular subsections was less clear than some of the other text in the paper. As a result, I came away with a few points of conclusion. For example, I remain confused about why exactly the issues that affect DIC and the carbon isotopes don't affect the oxygen isotopes. It would be helpful to spell this out, and to do some rewriting in these two sections to step through your argument more clearly.

**Figure 5, 6, 7, & 10**

There are big gaps in the stratigraphic data here. These gaps are mentioned obliquely in line 390, but not clearly laid out in section 3. Why are there such large gaps in sampling? These intervals seem pretty important to the interpretation of your data. Importantly, I don't think you need to have these data to make your point. Rather, I just think the communication is trying to gloss over the missing data. This is happening both in the writing and in the visual communication.

I think a more fair representation of the data completeness is needed. On the written side, would like to see this part of the data explicitly explained somewhere in section 3.1. In the figures, I think that including a dashed line connecting your data in these plots is misleading; it implies a continuity of data you don't have. These connecting lines should be eliminated.

This is especially important in figure 10, which actually does not have dashes. Rather, here, the missing data are represented by a solid, which implies *even more* certainty. This one especially should just have gaps in the record where you have big stratigraphic distances between sampling points.

**Age model in Figure 10**

It is not made clear how you establish an age model for figure 10. Is it just a linear interpolation based on the cron tie points? This makes some sense, and would be the most complicated procedure your data can justify, but I would like to see it explicitly described somewhere. Or did I miss it?

**Typos/Misspellings/Style**

I noticed several places where there were copyediting issues. For example: The legend of figure 1 says "Li**to**stratigraphy", not "Li**tho**stratigraphy" These will likely be mostly caught during copyediting, but since this is a quality article already, the authors should take the time this round of edits to go through the thing with a fine-toothed comb.

I also noticed a few instances where the authors could improve their style. For example:
1.  "starving" of what? I assume you mean oxygen, but it could mean food as well.          line 60

2. You use "key" twice in this sentence, which gives me the impression that you have a   line 52
whole keyring. Moreover, the actual meat of the sentence is a little vague, and the reader
comes away with no concrete idea of what you mean. Rather than declaring that the
MECO can teach us something about the Earth system, I would re-write this to just
specifically state *what* it can teach us.

3. "suffered" is a strange word here. It applies a bit too much humanity to the oceans, for   line 55
my taste.

4. The (2.4 My) in parentheses is redundant.   line 50

These are a few examples, and I would urge the authors to make sure their language is really good this
round of edits, because I think the paper is pretty close.

---

## Author Comment (AC1)

**Response and additions/changes performed following the Referee 1 (Eric Barefoot) comments: R1Cx.**

**R1C-1**. **Synopsis**
**Response:** Thank you for the general appreciation of our old manuscript (OMS) and the summary of the reported research. The concern pointed in the (2) "main findings": "it is not made clear why oxygen isotopes are not affected by these processes." was thoroughly addressed in the revised manuscript (RMS) and in the responses to the reviewer **Minor comments**.
**Added/Changed:** none.

**R1C-2**. **Overall Comments**
**Response:** Thank you for the very positive comments on our study of the MECO in a well-chosen field site.
**Added/Changed:** none.

**R1C-3**. **Minor Comments**; **comments on: Primary vs Diagenetic Signals & MECO isotopic record "**I appreciate… then follow up with evidence."
**Response:** Thank you for this comment. We agree that while our writing reflects how we have worked and thought initially, it does not ease the reading. We have followed the reviewer's suggestion and switched the order of these two sections.
**Added/Changed:**
Section 5.1 in OMS is Section 5.2 in RMS (lines 391 to 432).
Section 5.2 in OMS is Section 5.1 in RMS (lines 352 to 390).
Both sections were reorganized, several sentences were rewritten, and some explanations and references were added.

**R1C-4**. **Minor Comments; comments on: Primary vs Diagenetic Signals & MECO isotopic record** "Additionally, I found the language in these two particular subsections was less clear than some of the other text in the paper..."
**Response:** Yes, we agree. The discussion on the processes affecting the carbon and oxygen isotope composition ($\delta^{13}C$ and $\delta^{18}O$ values) of the dissolved inorganic carbon (DIC) and carbonates were reorganized and reworded in the RMS.
**Added/Changed:**
"5.2 Primary versus diagenetic signals
The carbonate primary carbon and oxygen isotope compositions may be affected by postdepositional processes, including the neoformation of authigenic and diagenetic phases. Therefore, before the paleoenvironmental interpretation of $\delta^{13}C_{carb}$ and $\delta^{18}O_{carb}$ records from shallow marine environments, it is necessary to determine primary versus diagenetic signal components. This discrimination requires understanding the factors controlling the primary marine isotopic composition and an evaluation of potential diagenetic overprints on the original geochemical signatures (e.g., Marshall, 1992; Schrag et al., 1995).
Oxygen isotopes in carbonates are controlled by the temperature of formation, the $\delta18O$ value of the carbonate-precipitating fluid ($\delta18O_w$), the mineralogy (e.g., higher $\delta^{18}O$ in dolomite vs. calcite), and any environmental parameter (e.g., pH, salinity) affecting the rate of carbonate precipitation (Swart, 2015). The effect of diagenetic alteration is more pronounced in the case of oxygen isotopes than carbon isotopes due to the high amount of oxygen relative to carbon present in postdepositional fluids and their variable $\delta^{18}O$ values (e.g., Marshall, 1992; Schrag et al., 1995; Fio et al., 2010). Carbonate with low $\delta^{18}O$ values can be produced by increasing temperature, freshwater input, and meteoric diagenesis, whereas $^{18}O$ enrichment could indicate either lower temperature or evaporation (e.g., Marshall, 1992; Patterson and Walter,1994; Schrag et al., 1995). In contrast, carbon isotopes are not thought to be directly influenced by temperature and are generally more resistant to diagenetic processes (Patterson and Walter,1994; Schrag et al., 1995; Swart, 2015). However, $\delta^{13}C$ values are also controlled by kinetic effects, mineralogy, and mainly by the $\delta^{13}C$ value from the DIC (Wendler, 2013). The primary diagenetic process that affects the

$\delta^{13}C$ values of the DIC is the oxidation of the organic matter, which produce $CO_2$ (and DIC species) depleted in $^{13}C$ (low $\delta^{13}C$ values). Therefore, the $\delta^{13}C$ values of the DIC and derived carbonates indicate the source of carbon, including the type of degraded/oxidized organic matter (OM) of different types, original seawater carbon, skeletal and non-skeletal carbonate sources (e.g., Swart, 2015). In proximal depositional environments, however, the δ13C values could be modified by (1) OM source, productivity, and burial rate, (2) extrabasinal carbonate input, (3) water circulation/stratification and evaporation, (4) terrestrial runoff and weathering (Saltzman and Thomas, 2012, Läuchli et al., 2021). Considering this, $\delta^{13}C$ is usually used as a global correlation tool since it can register eustatic sea-level fluctuations, changes in weathering flux, or significant perturbations in the global carbon cycle (e.g., volcanic $CO_2$ input; Wendler 2013 and references therein).

The degree of diagenetic alteration was assessed through three different approaches. First, was evaluated the relationship between $\delta^{13}C$ and $\delta^{18}O$ values (Brasier *et al*., 1996). Statistically, a non-significant correlation (Pearson correlation coefficient; $r < 0.6$) indicates that a diagenetic overprint of the primary isotopic signature can be excluded (e.g., Fio *et al*., 2010). In both sections, no statistical significant correlation ($r < 0.3$) was found between the $\delta^{18}O_{carb}$ and $\delta^{13}C_{carb}$ values. This lack of relationship suggests that no or minor diagenetic modifications affected the primary isotopic compositions (Fig. 10). The second approach used to assess the degree of alteration uses clay mineralogy. Kübler and Jaboyedoff (2000) defined four diagenetic zones by comparing illite crystallinity with mineral assemblages and organic matter type. The Belsué and Yebra de Basa samples have 20–30% smectite within the illite-smectite (IS) mixed layers and are within the 3rd diagenetic zone of Kübler and Jabeyedoff (2000), i.e., shallow diagenesis (*ca.* 60–80°C). Another diagenetic indicator is the maximum temperature ($T_{max}$) reached during the Rock-Eval Pyrolysis (S2), which marks the maturity of the OM. The $T_{max}$ values obtained in samples with relatively high OM content (TOC > 0.5 wt.%; S2 > 0.2) were < 440°C (Fig. 8), which corresponds to the beginning of the oil window (*ca.* 60°C; Espitalié et al., 1985). This maturity level of the organic matter agrees with vitrinite reflectance and Raman measurements in the studied area (Labaume et al., 2016). In summary, the three approaches for assessment of the diagenetic degree, i.e., carbonate $\delta^{13}C$ and $\delta^{18}O$ values, illite crystallinity, and thermal maturation of the organic matter ($T_{max}$), suggest that the diagenetic overprint in the studied Belsué and Yebra de Basa rocks is low. The primary isotopic signal is preserved largely in both sections. It can be safely used to study paleoenvironmental conditions and be compared to global key isotopic curves during the MECO event."

**R1C-2.** "Figure 5, 6, 7, & 10..."

**Response:**
Agree.
There are four main gaps in data: 1) on Belsué-E section between ~55 and ~65 m, 2) on Belsué-E section between ~85 and ~115 m, and 3) on Yebra de Basa (HR, figure 4) section between ~100 and ~120 m, and 4) on Yebra de Basa (HR, figure 4) section between ~180 m and ~200 m.

In section 3.1 "**A total of 101 samples in BS and 157 samples in YB were collected, each of them was composed by ca. 200 g of fine-grained and fresh rock from below the weathering depth to avoid alteration and grain size bias**", we did not emphasize enough that we tried to sample in the most carbonate rich and as homogeneous as possible fine-grained material. This corresponds to the marls. They represent similarly deep environments, and are carbonate, organic and clay rich, which is what the type of proxies we required. Sampling the sandy clastic intervals can be performed when one only looks at the organic matter, but is less than ideal to explore primary signals in carbonates. Although the exposure conditions are usually ideal for this work, difficulties in sampling in this field area can arise because of either 1) a dominance of sandy facies at the outcrop, or 2) insufficient exposure due to the fine-grained nature of marls (marly intervals in steep topography are usually providing excellent outcrops, but if situated in topographic depressions they can also be more vegetated and lacking exposure).

Samples free intervals 1, 3 and 4 correspond to the most sandy intervals at the moments of maximum deltaic progradation. Sample free interval 2 results of both coarse-grained outcrops at the level of the second progradation in Belsué, and of lack of sufficient exposure in the marlier interval above this progradation. To be more fair in the representation, we highlighted the poor exposure zones and data gaps in Figure 4. Moreover, we erased in Figures 5,6,7, and 10 the line that connect the different intervals of the four data gaps, and we added a fine dashed line of light grey to differentiate the sample and non-sample intervals.

It remains important to note that, as explained by reviewer 1, given our magnetostratigraphic constraints and the fit with global curves, the absence of data in these three intervals has fortunately no impact on our results/conclusions.

**Added/Changed:**

Added, line 164 to 165 in the RMS: [**The samples were mostly marls, corresponding to rocks rich in carbonate, OM, and clays.**]

Added, lines 168 to 171 in the RMS: [**The exposure conditions were usually ideal for sampling in both sections. However, there were difficulties in four intervals, resulting in gaps in the data. The problems were due to a dominance in sandy facies at the outcrop, corresponding to moments of maximum deltaic progradation, or to poor exposure because of the fine-grained nature of the marls (e.g., Quaternary cover).**]

Added, lines 252 to 256 in the RMS: [**Three of these data gaps were due to the dominance of sandy facies. In YB, the sandy intervals correspond to the Sabiñánigo sandstone deltaic bodies located approximately at 100–120 m and 180–200 m (YB-HR section; Fig. 4). In BS, the Belsué sandstone interval is placed between 55 and 60 m (Belsué-E section; Fig. 4). The fourth data gap located at 85–115 m in Belsué-E, results of lack of sufficient exposure within the marls and the presence of a coarse-grained sandy interval (Fig. 4).**].

Change in figure 4: **Highlighted poor exposure zones**

Change in figures 5, 6, 7, and 10: **Erased connecting lines of data gaps, and added a fine dashed line of light grey to differentiate the sample and non-sample intervals.**

**R1C-3.** "Age model in Figure 10..."

**Response:**
Agree. This is missing. Indeed, we simply scaled our sections and the corresponding proxy records based on the magnetostratigraphic tie points (with repositioning our data on Garcés et al., 2014 and Vinyoles et al., 2021 magnetostratigraphic sections).

**Added/Changed:**

Added in the legend of figure 9 (RMS), lines 366 to 367: [Figure 9: Oxygen isotope ($\delta^{18}O_{carb}$) correlation panel for the studied sections (Belsué and Yebra de Basa) with MECO target curves from Alano (Italy, Tethys Ocean, Spofforth et al., 2010), ODPS 1051 (N Atlantic Ocean; Edgar et al., 2010), ODPS 702 (S Atlantic Ocean; Bohaty et al., 2009) and ODPS 738 (S Indic Ocean; Bohaty et al., 2009). Data from the bulk and fine sediments fractions. Highlighted in red the OM rich interval (TOC peak) in Yebra de Basa. The two progradation-retrogradation cycles referred in the text are drawn with grey and white triangles. **The data are scaled according to magnetostratigraphic tie points between C18r-18n.2n and C18n.2n-C18n.1r chrons.**]

*Typos/Misspelling/Style*

**R1C-4.** "I noticed several places where there were copyediting issues. For example: The legend of figure 1 says "Litostratigraphy", not "Lithostratigraphy"..."

**Response:** Thanks for pointing out these.

**Added/Changed:**

Corrected **Figure 2. We changed "**Litostratigraphy**" by "Lithostratigraphy" and "**Depositonal**" by "Depositional".**

We double-checked the RMS and have corrected typos, grammatical mistakes, and bad choose of words. The English was revised.

**R1C-5.** "I also noticed a few instances where the authors could improve their style. For example:

1. "starving" of what? I assume you mean oxygen, but it could mean food as well. line 60..."

**Response:** Corrected.

**Added/Changed:**

Lines 61 to 65 in the RMS:

We added new information and changed the final sentence: [**However, while the temperature increase in the oceans has been inferred in multiple sites, the MECO environmental perturbation affected differently the fauna communities (Arimoto *et al*., 2020). In some locations, the warmer conditions reduced nutrient availability, decreasing the benthic productivity (Arimoto *et al*., 2020; Bijl *et al*., 2010, Galazzo *et al*., 2014; Moebius *et al*., 2015). In contrast, the Southern Ocean (Moebius *et al*., 2014) or the Neo-Tethys Ocean (Galazzo *et al*., 2013) record increased productivity during the MECO.** ]

The style of the text was improved and several sentences were reworded, which can be easily find in the ms version with "track-changes".

**R1C-6.** "2. You use "key" twice in this sentence, which gives me the impression that you have awhole keyring. Moreover, the actual meat of the sentence is a little vague, and the reader comes away with no concrete idea of what you mean. Rather than declaring that the MECO can teach us something about the Earth system, I would re-write this to just specifically state what it can teach us. line 52..."

**Response:** Agree.

**Added/Changed:**

Lines 50 to 55 in RMS:

**We modified the "key" elements and provide additional text to explain more specifically why we care about the MECO:**

[**Therefore, considering the unresolved MECO driving mechanism(s), and how the Earth system responded to this carbon cycle perturbation, the MECO poses a significant challenge to understanding carbon cycle variations on timescales of several hundreds of thousands of years (Sluijs *et al*., 2013; Henehan *et al*., 2020; Sternai *et al*., 2020). Addressing this challenge requires extensive documentation of the MECO in a range of environments and geodynamic contexts, as well as documentation of its effect on Earth surface dynamics.**]

**R1C-7.** "suffered" is a strange word here. It applies a bit too much humanity to the oceans, for my taste. line 55"

**Response**: Corrected

**Added/Changed:** Line 56: **We changed "suffered" by "experienced".**

**R1C-8.** "The (2.4 My) in parentheses is redundant. line 50"

**Response**: Corrected

**Added/Changed:**  Line 50: **We erased the parentheses**

---

## Author Comment (AC2)

**Referee 2 (Anonymous) comments: R2C2**

*Synopsis*

**R2C-1**. "section 5. 2. The major issue concerns the lack of interpretations for d18O fluctuations (and to a lesser extent this comment can be also applied to the d13Ccarb). Most of the time, this section only contains descriptions of isotopic trends without giving quantitative estimates about the MECO deduced from data presented in the section 4."

**Response**:

Thank you for this observation. In order to find the possible correspondence with global target curves and identify the MECO, in this paper we focused on describing the trends. It is true we lack quantitative estimates, but the small-scale fluctuations of $\delta^{18}O_{carb}$ could be linked to many factors that we have no control on. Contrary to the MECO isotopic excursion that is well characterized and for which, we also have magnetostratigraphic constraints.

However, in the interpretation part of the OMS, we come back on the link between the MECO $\delta^{18}O_{carb}$ fluctuations and the prominent deltaic progradation observed on both sections, hence addressing the relationship between the climatic perturbation, surface processes, and sediment supply to the basin.

**Added/Changed:** The section 5.1 and 5.2 were reorganized and several sentences reworded.

**R2C-2**. "lines 467-469 and 497-499. To my understanding, one of the main findings of the MS seems to be the significant increase of erosion and sediment transport during the MECO. However the figure 11 clearly shows a sedimentation rate held constant through time (pre/syn/post-MECO). This point deserves more attention."

**Response**:

We thank the reviewer for raising this point, which is important, and it wasn't sufficiently addressed in the OMS. First, the sedimentation rate (SR) indicated on figure 11 are "average" SR taken from Vinyoles et al. 2021. Indeed, these authors do not have sufficient time resolution to deconvolve high-resolution SR variations, and neither do we. In essence, there are no data here that would allow us to discuss variations in SR at the scale of the observed variations in $\delta^{18}O$.

On the other hand, in such a shallow marine environment, the SR is not necessarily a faithful indicator of increased erosion in sediment transport. Indeed, with limited space for accommodation, an increase in sediment supply is in fact more expressed by progradation and sediment bypass than by an increase in SR. We hence focus on sequence stratigraphic interpretation (A/S ratio) as a proxy of variations of sediment supply and accommodation. In theory, a test of our suggestion would be to assess sediment volumes, but this is currently impossible given the incompleteness of the outcrops in 3D at this scale.

**Added/Changed:** The section "5.4 MECO response in the South Pyrenean Foreland Basin" was

was checked and edited when necessary in the RMS (Lines 471–506).

**R2C-3**. "Lines 500-501. A such increase of the hydrological cycle during the MECO should, by its action on the weathering (and the alkalinity delivered), inhibit the ocean acidification process (or the authors assume that the acidification takes place in the early stage of the MECO before the enhancement of the hydrological cycle ?). Consequently, the authors should reword some of their points mentioned as potential implications (section 5.5) to be consistent with their results."

**Response**:

Thank you for this suggestion. The ocean acidification observed during the MECO is indeed controversial, as raised by Sluijs et al., (2013). They dubbed the MECO as "a middle Eocene carbon cycle conundrum", since the observations do not fit with the current carbon cycle theory.

**Added/Changed:**

To be more consistent with the results and not link ocean acidification and enhanced hydrological cycle, we have modified lines 527–529 of the RMS: **[This response in sediment delivery rate, OM burial in shallow and restricted basins, has been previously documented for other early Eocene hyperthermals (Chen *et al*., 2018; Foreman *et al*., 2012, 2014; Pujalte *et al*., 2015; Foreman and Straub, 2017; Honegger *et al*., 2020).]**

**R2C-4**. "Figure 3 (caption) Where are the grey lines mentioned in the caption?"

**Response**:

Thank you for this observation. The grey lines mentioned in the caption define the four depositional sequences defined by Millán et al., 1994 in the Belsué syncline.

**Added/Changed:**

Figure 3 was improved, we have emphasized the lines representing the depositional sequencies with a darker, more visible, grey.

**R2C-5**. "lines 366 and 368: please do not mix r and $r^2$ "

**Response**:

Corrected. Text and figures were checked and changed when necessary. In the RMS is used consistently the Pearson correlation coefficient, r.

**Added/Changed:** We modified in the RMS:

Lines 428–499: " Statistically, a non-significant correlation (**Pearson correlation coefficient;** *r* **< 0.6**) indicates that a diagenetic overprint of the primary isotopic signature can be excluded (e.g., Fio *et al*., 2010). In both sections, no statistical significant correlation (*r* **< 0.3**) was found …"

Lines 460 to 463: " In addition, the terrestrial origin is also supported by the strong correlation (**r > 0.7**) observed between the siliciclastic elements (Al, Ti, Fe) and the TOC or all the OM-related trace elements (V, Mo, Ba, and Th; Tribovillard *et al*., 2006).